# Long-term stability and the physical and chemical factors predictive for antimicrobial activity in Australian honey

Kenya E. Fernandes[1], Andrew Z. Dong[1], Aviva Levina[2], Nural N. Cokcetin[1], Peter Brooks[3], Dee A. Carter[1,4]*

1 School of Life and Environmental Sciences, University of Sydney, Sydney, New South Wales, Australia, 2 School of Chemistry, University of Sydney, Sydney, New South Wales, Australia, 3 School of Science, Technology and Engineering, University of the Sunshine Coast, Maroochydore, Queensland, Australia, 4 Sydney Institute for Infectious Diseases, University of Sydney, Sydney, New South Wales, Australia

* dee.carter@sydney.edu.au

**Data Availability Statement:** All relevant data are within the manuscript and its Supporting information files.

## Abstract

The growing burden of expired medicines contributes to environmental contamination and landfill waste accumulation. Medicinal honey, with its non-toxic nature and potentially long shelf-life, represents a promising and underutilised therapeutic that avoids some of these issues. However, limited knowledge on how its antimicrobial properties change over time combined with a lack of reliable processes in the honey industry for measuring antimicrobial potential, hinder its clinical adoption. Using a diverse selection of 30 Australian honey samples collected between 2005 and 2007, we comprehensively evaluated their antibacterial and antifungal activity and pertinent physical and chemical properties with the aims of assessing the effect of long-term storage on activity, pinpointing factors associated with antimicrobial efficacy, and establishing robust assessment methods. Minimum inhibitory concentration (MIC) assays proved superior to the standard phenol equivalence assay in capturing the full range of antimicrobial activity present in honey. Correlations between activity and a range of physical and chemical properties uncovered significant associations, with hydrogen peroxide, antioxidant content, and water activity emerging as key indicators in non-*Leptospermum* honey. However, the complex nature and the diverse composition of honey samples precludes the use of high-throughput chemical tests for accurately assessing this activity, and direct assessment using live microorganisms remains the most economical and reliable method. We provide recommendations for different methods of assaying various honey properties, taking into account their accuracy along with technical difficulty and safety considerations. All *Leptospermum* and fourteen of seventeen non-*Leptospermum* honey samples retained at least some antimicrobial properties after 15–17 years of storage, suggesting that honey can remain active for extended periods. Overall, the results of this study will help industry meet the growing demand for high-quality, medicinally active honey while ensuring accurate assessment of its antimicrobial potential.

**Funding:** Funding for this project was provided by NSW Bushfire Industry Recovery Package Sector Development Grant (BIP-SDG-135) to DC. The funders had no role in study design, data collection and analysis, decision to publish, or preparation of the manuscript.

**Competing interests:** The authors have declared that no competing interests exist.

## Introduction

All medicines have a shelf life, and the fate of expired medicines is an area of increasing concern. Despite the free returned medicine scheme available in Australia, many medicines are disposed of inappropriately or retained in the household well beyond their expiry date [1]. These issues are of particular concern when it comes to antimicrobials, where inappropriate disposal can result in environmental contamination [2] and use outside of prescribed need can lead to poor clinical outcomes and the development of antimicrobial resistance [3]. Natural products with antimicrobial activity could mitigate such risks, with a promising option being medicinal honey. As a non-toxic product, medicinal does not pose an environmental risk, and it appears incapable of inducing resistance [4, 5]. Additionally, when stored correctly, honey boasts an exceptionally long shelf life. A notable example is the discovery of edible honey in the tomb of Egyptian king Tutankhamun, which remained preserved for over 3,000 years [6].

Despite growing recognition of the potential therapeutic benefits of honey, there remains a need for evidence-based data to promote its use in clinical practice. Linked to this a desire in the medicinal honey industry to identify key factors that can determine whether a given honey will be antimicrobial, and to devise reliable processes for measuring these factors. Honey has an extremely complex chemical composition consisting of sugars, organic acids, enzymes, and bioactive compounds like flavonoids and phenolic acids [7], and these can all vary substantially with floral source, geographic location, processing methods, and colony health [8, 9]. These components are susceptible to changes over time, with storage conditions playing an important role in the stability of key properties [10–12]. The absence of well-defined parameters for assessing antimicrobial activity, coupled with limited data on the impact of long-term storage on activity, presents a substantial challenge to the honey industry and ultimately affects the clinical uptake of honey. Addressing these knowledge gaps is essential to meet the demand for high-quality medicinally active honey and to maximise its therapeutic potential.

Antimicrobial activity can be measured in honey using various methods, each with its advantages and limitations [13]. In Australia, the prevalent approaches are the phenol equivalence assay and the minimum inhibitory concentration (MIC) assay. The phenol equivalence assay, the current Australian industry standard, is an agar diffusion assay that compares the activity of honey against *Staphylococcus aureus* with a standard phenol solution [14]. The MIC assay, a standardised research method, is a broth dilution assay that determines the minimum concentration of honey required to inhibit microbial growth [15]. Antimicrobial activity can also be estimated through the measurement of certain chemical properties. In honey derived from *Leptospermum* nectar, most antibacterial activity is due to the presence of methylglyoxal (MGO) and the two correlate very well, allowing a simple chemical test of MGO concentration to replace biological testing [16]. The major antimicrobial factor in non-*Leptospermum* honeys is thought to be hydrogen peroxide ($H_2O_2$) production, which can be measured in honey using various methods including the horseradish peroxidase (HRP) assay [17] and AmplexRed-based commercial kits [18]. However, these tend to be either technically difficult or costly to perform and often correlate poorly with antimicrobial levels, suggesting interference by other honey components [17, 19, 20].

In an extensive survey of Australian honey, Irish et al. (2011) characterised 477 samples for their antimicrobial activity using the phenol equivalence assay [12], and assessed the stability of activity of a selected subset following short-term (8 to 22 months) storage. In this follow-up study, we chose a diverse selection of 30 honey samples from a more restricted geographic area, with the aim of assessing the effect of ~15 years of storage on antimicrobial activity. Finding most honeys retained a useful level of antimicrobial activity, this collection then allowed us to explore a range of parameters of interest to the honey industry, including antibacterial and

antifungal activity, hydrogen peroxide production, phenolic and antioxidant content, methyl-glyoxal content, and honey colour. Where possible, we compared different ways of measuring honey properties to determine the most useful and reliable methods. Finally, we combined the data to examine the relationships between the antimicrobial activity of honey and its physical and chemical properties, thereby identifying the key factors that contribute to its therapeutic potential.

## Materials & methods

### Honey samples and preparation

Honey samples previously screened by Irish et al. between 2005 and 2007 were chosen for analysis [12]. Since their original assessment, all honeys have been stored in plastic containers in the dark at 4 ˚C. Fifteen paired honey samples were selected, including native, non-native, *Leptospermum*, and non-*Leptospermum* floral sources and with no, low, medium, and high antimicrobial activity (S1 Table). The floral source of the honey was determined by beekeepers through a process that considered the floral resources available for nectar collection, the geographical location of the apiary, and organoleptic characteristics of the honey. Three control honeys were included in all tests: artificial honey (1.5 g sucrose, 7.5 g maltose, 40.5 g fructose, 33.5 g glucose, 17 mL sterile water) (ART), Barnes TA 10+ Jarrah honey (JAR) with peroxide-based activity, and Comvita UMF 18+ Manuka honey (MAN) with non-peroxide-based activity. Unless otherwise specified, all honey samples were mixed thoroughly with a spatula, incubated at 35 ˚C for 15 minutes, diluted to the target concentration in sterile water, and vortexed thoroughly before use.

### Microorganisms and culture conditions

Two Gram-positive bacterial species (*Staphylococcus aureus* ATCC29213, *Enterococcus faecalis* ATCC29212), one Gram-negative bacterium (*Pseudomonas aeruginosa* ATCC29883), and one yeast (*Cryptococcus deuterogattii* R265) were used for antimicrobial testing. Bacterial and yeast strains were maintained as glycerol stocks at –80 ˚C. Bacterial strains were grown on Nutrient Agar (NA; Oxoid) and incubated at 30 ˚C for 24 hours before use. *C. deuterogattii* was grown on Potato Dextrose Agar (PDA; Oxoid) and incubated at 30 ˚C for 48 hours.

### Antimicrobial susceptibility testing

The phenol equivalence assay was performed based on the method outlined in Irish et al. 2011 [12]. Antimicrobial susceptibility testing by broth microdilution was performed in accordance with CLSI guidelines for aerobic bacteria [21] and yeasts [22] with minor modifications. Briefly, inocula were prepared from colonies growing on agar plates to a final concentration of $2 \times 10^5$–$8 \times 10^5$ for bacteria, and $0.5 \times 10^3$–$2.5 \times 10^3$ CFU/ml for *C. deuterogattii*. Bacterial strains were adjusted to an absorbance of between 0.08–0.1 at 540 nm while *C. deuterogattii* cells were counted using a haemocytometer. Assays used Mueller-Hinton Broth (MHB; Oxoid) supplemented with 20 mg $Ca^{++}$/L and 10 mg $Mg^{++}$/L for bacteria, and Yeast Nutrient Broth (YNB; Sigma-Aldrich) supplemented with 0.165 M MOPS and 0.5% D-glucose for *C. deuterogattii*. Honeys were assayed at 30, 25, 20, 15, 10, and 5% (w/w) diluted in either sterile water or freshly prepared 5600 U/ml catalase solution, and control drugs were assayed at concentration ranges of 0.0039–4 µg/ml for amphotericin B (AMB) and 0.125–32 µg/ml for tetracycline (TET). Plates were incubated without agitation at 35 ˚C for 18 hours (bacteria), or 48 hours (*C. deuterogattii*). The MIC was determined visually and defined as the lowest drug/

honey concentration at which growth was completely inhibited (no visible turbidity). Three independent repeats were performed for each honey sample.

## Honey microbial content

After thorough mixing, 100 μl of undiluted honey was placed on either NA or V8-Juice Agar (Oxoid) using a truncated P1000 pipette tip. A sterile cotton swab was used to spread the honey evenly across the entire surface of the agar plate. Plates were incubated at 35 ˚C for 5 days and photographed using a digital camera. Two independent replicates were performed on different days.

## Hydrogen peroxide assays (HRP, ADHP, and strip-test)

The horseradish peroxidase (HRP) assay was performed based on the method outlined in Lehmann et al. 2019 [17]. For the 10-acetyl-3,7-dihydroxyphenoxazine (ADHP) assay, honey samples were diluted to 25% (w/w) in sterile water. At the 2-hour timepoint, 40 μl of sample was aliquoted into a 96-well plate, 160 μl of ADHP reagent (1 mM ADHP and 30 μg/ml HRP in PBS) was added and absorbance was measured at 560 nm using a plate reader. For both HRP and ADHP assays, 2-fold dilutions of $H_2O_2$ standards ranging from 0.5–1024 μM were used to generate standard curves and the resulting equations from the lines of best fit were used to calculate the amount of $H_2O_2$ in each honey sample. For the strip-test method, honey samples were diluted to 25% (w/w) in sterile water. At the 2-hour timepoint, $H_2O_2$ test strips were immersed in honey samples for 1 second and the excess shaken off. Test strips were allowed to develop for 10 seconds and compared to the colour chart provided by the manufacturer to determine $H_2O_2$ concentration.

## Phenolics assays (Folin-Ciocalteu and fast blue BB)

For the Folin-Ciocalteu (FC) assay, 20 μl aliquots of 20% (w/w) honey samples were prepared in triplicate in a 96-well plate. To each sample, 100 μl of FC reagent (1 ml Folin-Ciocalteu reagent in 30 ml sterile water) was added with incubation in the dark for 5 min at RT. Next, 80 μl of 0.75% $Na_2CO_3$ was added with incubation in the dark for 90 min at RT. Absorbance was measured at 760 nm using a plate reader. The Fast Blue BB (FBBB) assay was performed based on the method outlined previously in [23], with slight modifications. In a 96-well plate, 200 μl aliquots of 20% (w/w) honey samples were prepared in triplicate. To each sample, 20 μl of 0.1% Fast Blue BB reagent was added and pipetted up and down vigorously for 30 sec. Next, 20 μl of 5% NaOH was added with incubation in the dark for 45 min at RT. Absorbance was measured at 420 nm using a plate reader. For both assays, gallic acid standards ranging from 0.06–0.18 mg/ml were used to generate a standard curve and the resulting equation for the line of best fit was used to calculate the phenolics content of honey samples, expressed as mg of gallic acid equivalent per kg of honey (mg GAE/kg).

## Antioxidant assays (FRAP & DDPH)

For the ferric-reducing antioxidant power (FRAP) assay, FRAP reagent was prepared freshly from 1:1:10 (v/v/v) of 10 mM 2,4,6-Tris(2-pyridyl)-s-triazine in 40 mM HCl, 20 mM $FeCl_3$, and 300 mM pH 3.6 acetate buffer, and was warmed to 37 ˚C prior to the assay. In a 96-well plate, 20 μl aliquots of 20% (w/w) honey samples were prepared in triplicate. To each sample, 180 μl of FRAP reagent was added with incubation for 30 min at 37 ˚C. Absorbance was measured at 594 nm using a plate reader. $FeSO_4$ standards ranging from 200–1200 μM, made freshly and stored on ice until use, were used to generate a standard curve and the resulting

equation from the line of best fit was used to calculate FRAP value, expressed as μmol $Fe^{2+}$/kg. For the 2,2-diphenyl-1-picrylhydrazyl (DPPH) assay, 10 μl aliquots of 20% (w/w) honey samples were prepared in triplicate in a 96-well plate. To each sample, 100 μl of 100 mM pH 5.5 acetate buffer and 250 μl of DPPH reagent (130 μM DPPH in methanol) were added with incubation in the dark for 90 min at RT. Absorbance was measured at 520 nm using a plate reader against a methanol blank. Trolox standards at pH 7 ranging from 100–600 μM were used to generate a standard curve and the resulting equation from the line of best fit was used to calculate radical scavenging activity, expressed as μmol Trolox equivalent per kg of honey (μmol TE/kg).

## Assessment of honey colour

For visual assessment, 10 ml of undiluted honey in a 15 ml Falcon tube was compared against a Pfund colour grader chart. For spectrophotometric assessment, honey samples were diluted to 50% (w/w) and a wave scan from 380–780 nm at 1 nm steps was taken using a UV/Vis spectrophotometer (Specord S600) in disposable plastic cuvettes with 10 mm optical pathlength. Sterile water was used as a blank. Honey colour was then assessed using three methods: Pfund value, colour intensity, and CIELAB coordinates. Pfund values were calculated using the equation developed by White in 1984 [24], $Pfund\ (mm) = -38.70 + 371.39(A_{635})$. Colour Intensity was calculated using the equation $Colour\ Intensity\ (mAU) = (A_{720} - A_{450}) \times 1000$. CIELAB coordinates were calculated using the method outlined in [25].

## Assessment of other physical and chemical properties

For pH measurements, 10% (w/w) honey samples were measured using a pH meter (Mettler Toledo) at RT. Brix value was measured at 20 ˚C using a refractometer (Hanna HI96801), with Brix value expressed as g of total sugar per 100 g of honey and water content expressed as % (w/w). Water activity was measured at 25 ˚C using a water activity analyser (Aqualab PRE) with a correction of ± 0.005 $a_w$ made per 0.1 ºC deviation. For determination of methylglyoxal, dihydroxyacetone, and hydroxymethylfurfural content, honey samples were derivatised with O-(2,3,4,5,6-pentafluorobenzyl)hydroxylamine HCl and analysed against Anisole by HPLC with 263 nm detection, as per [26].

## Elemental content

Honey samples were digested using a previously described method [27] with modifications. An approx. 9 M solution of $HNO_3$ was prepared by mixing two volumes of Milli-Q water with three volumes of 65% $HNO_3$. Honey samples were heated at 40 ˚C to increase fluidity, and three aliquots of each sample (~0.4g, ± 0.2 mg precision) were weighted into 10 mL polypropylene tubes. Samples were mixed with 4 mL of $HNO_3$ solution, heated up to 60 ˚C for 24 h, cooled to room temperature, and the sample volumes brought to 5 mL with Milli-Q water. To validate the digestion technique, selected samples were also digested using a harsher method taken from [28]. Weighed honey aliquots (~0.5 g) were mixed with 1 mL of Milli-Q water and dissolved by sonication at 50 ˚C. Then, 5 mL of 65% $HNO_3$ was added to each sample, and the samples were heated for 1 h at 90 ˚C (caution: this results in a violent reaction with the formation of $NO_2$ gas). Samples were cooled, and 1 mL of 30% $H_2O_2$ was added. After further heating for 30 min at 90 ˚C, the samples were cooled to room temperature and brought to 10 mL with Milli-Q water. The elemental contents measured using both digestion techniques agreed with 5% experimental error. Elemental analysis of the digests was performed on a Perkin-Elmer NexION 350X ICP-MS analyser. A 24-element balanced environmental standard (HPS-Q0109521 from High Purity Standards) was used for calibration. Typical calibration

ranges were 10–10,000 ppb for highly abundant elements and 0.10–100 ppb for trace metals. Only the elements that were consistently at least twice as high in honey samples compared to the blanks were included in the final analysis.

## Statistical analysis & figure preparation

Whether data were normally distributed was determined using the Shapiro-Wilk normality test. For statistical analysis, MIC readings above the tested maximum of 30% (w/w) honey were assigned a value of 35%. Associations between variables used Spearman's rank correlations which determines the strength and direction of monotonic relationships but does not assume linearity, or linear regressions which do assume linearity. Correlations between antimicrobial activity and the physical and chemical properties of honey samples were visualised using partial least squares (PLS) regression to identify the underlying relationships between the two sets of variables. For all correlations between antimicrobial activity and the physical and chemical properties of honey, MIC values were subtracted from 35 to reverse their direction so that variables related with increased antimicrobial activity would appear as positively rather than negatively correlated. P values <0.05 were considered significant. Data were analysed using Prism 5 (GraphPad Inc), and XLSTAT (Lumivero) software.

## Results

### Most honey samples retained some activity following 15–17 years of storage

The antimicrobial activity of honey samples was first assessed using the phenol equivalence (PE) assay and compared to the results from the original testing that took place between 2005–2007 (Fig 1A, with 2022 results also shown in Table 1). Most honey samples retained some level of activity after 15–17 years of storage, with fourteen out of seventeen non-*Leptospermum* honeys and four out of four *Leptospermum* honeys remaining active. The nine honey samples that were inactive when screened in 2005–2007 were still inactive. In the active non-*Leptospermum* honeys, the average total activity (TA) decreased by 64% from 15.6 to 5.5 PE, with a range of 3.5 to 25.1 PE. In the active *Leptospermum* honeys, the average non-peroxide activity (NPA) decreased by 31% from 17.0 to 11.9 PE, but this ranged from a 0.4 PE increase to an 8.6 PE decrease (Fig 1B).

The number of honey samples falling into the categories of no (<5), low (5–10), medium (10–20), and high (>20) activity at both testing times are shown in Fig 1C, with the largest category in 2005–2007 being medium (10–20) activity, and the largest category in 2022 being low (5–10) activity. Considering non-*Leptospermum* honeys only, a strong correlation was seen between the TA results from 2005–2007 and the results from 2022 (Spearman's rho = 0.87, p<0.001), and considering *Leptospermum* honeys only, a strong correlation was seen between NPA results (Spearman's rho = 0.96, p = 0.003). This indicates that TA decreased at a similar rate in non-*Leptospermum* honeys and NPA decreased at a similar rate in *Leptospermum* honeys, though the steeper TA curve indicates a faster decrease in TA relative to NPA.

### Minimum inhibitory concentration values generally aligned across species and with phenol equivalence results

The antimicrobial activity of honey samples was further assessed using broth microdilution methodology. *Staphylococcus aureus* was tested to compare results with the phenol equivalence assay, and additional microbes were tested including another Gram-positive (*Enterococcus faecalis*), a Gram-negative (*Pseudomonas aeruginosa*), and a fungus (*Cryptococcus deuterogattii*)

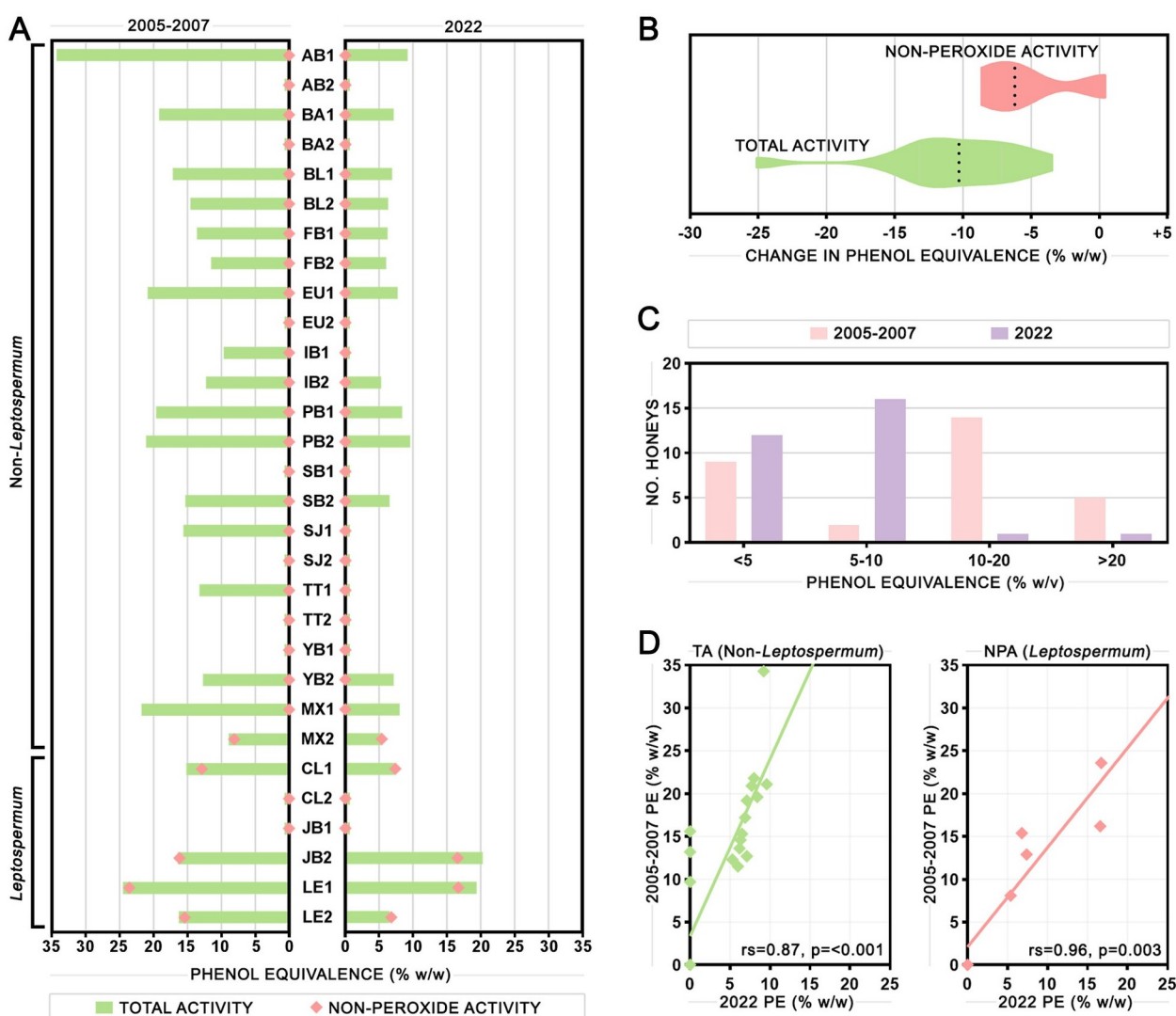

**Fig 1. Antimicrobial activity of honey samples over time measured by the phenol equivalence (PE) assay.** (A) Results of the PE assay performed in 2005–2007 (left) compared to 2022 (right) showing total activity (TA) and non-peroxide activity (NPA). (B) Box plots showing the distribution of change in phenol equivalence values for TA and NPA between 2005–2007 and 2022. Dotted line shows the median value. (C) Bar charts showing the number of honey samples in each activity range in 2005–2007 and 2022. (D) Correlations of TA between 2005–2007 and 2022 in non-*Leptospermum* samples only (left) and NPA between 2005–2007 and 2022 in *Leptospermum* samples only (right). Lines of best fit, Spearman's rho, and p-values are shown.

to see how consistent activity was against diverse microbes (Fig 2A and Table 1). Artificial honey MICs showed that *P. aeruginosa* is more susceptible to osmolarity than the other microbes tested with an MIC of 25% (w/w) compared to >30% (w/w) for the others. Overall, *S. aureus* was the most susceptible microbe to the TA of honey with an average MIC for active non-*Leptospermum* honeys of 14% (w/w), followed by *P. aeruginosa* at 16% (w/w), *C. deutero-gattii* at 22% (w/w), and *E. faecalis* at 26% (w/w). Considering NPA in *Leptospermum* honeys, *S. aureus* was again the most susceptible with an average MIC of 9% (w/w), followed by *E. fae-calis* at 16% (w/w), *P. aeruginosa* at 20% (w/w), and *C. deuterogattii* at 25% (w/w). Fig 2B shows the correlations between phenol equivalence and MIC values for TA and NPA in active honeys only for *S. aureus*. Considering TA, phenol equivalence and MIC values were weakly

**Table 1. Floral sources and antimicrobial susceptibility testing data for honey samples[1].**

| ID | Floral Source | S. aureus | | | | E. faecalis | | P. aeruginosa | | C. deutero-gattii | |
|---|---|---|---|---|---|---|---|---|---|---|---|
| | | PE 2022 | | MIC | | | | | | | |
| | | TA (% w/w) | NPA (% w/w) | TA (% w/w) | NPA (% w/w) | TA (% w/w) | NPA (% w/w) | TA (% w/w) | NPA (% w/w) | TA (% w/w) | NPA (% w/w) |
| *Non-Leptospermum Honeys* | | | | | | | | | | | |
| AB1 | Apple Box/Red Stringybark | 9.2 | 0.0 | 5 | >30 | 20 | >30 | 10 | 25 | 5 | 15 |
| AB2 | Apple Box | 0.0 | 0.0 | >30 | >30 | >30 | >30 | 25 | 25 | >30 | >30 |
| BA1 | Banksia | 7.1 | 0.0 | 10 | >30 | 30 | >30 | 15 | 25 | 20 | >30 |
| BA2 | Banksia | 0.0 | 0.0 | >30 | >30 | >30 | >30 | 25 | 25 | 30 | 30 |
| BL1 | Blueweed/ Lucerne | 6.9 | 0.0 | 20 | >30 | >30 | >30 | 20 | 25 | 30 | >30 |
| BL2 | Blueweed/ Lucerne | 6.3 | 0.0 | 20 | >30 | >30 | >30 | 20 | 25 | 30 | >30 |
| FB1 | Feather Bush | 6.2 | 0.0 | 10 | >30 | 20 | 25 | 15 | 25 | 30 | >30 |
| FB2 | Feather Bush | 6.0 | 0.0 | 10 | >30 | 20 | 25 | 15 | 25 | 30 | >30 |
| EU1 | Mixed Eucalypt | 7.7 | 0.0 | 5 | >30 | 20 | >30 | 10 | 25 | 10 | >30 |
| EU2 | Mixed Eucalypt | 0.0 | 0.0 | >30 | >30 | >30 | >30 | 25 | 25 | >30 | >30 |
| IB1 | Mugga Ironbark | 0.0 | 0.0 | >30 | >30 | >30 | >30 | 25 | 25 | >30 | >30 |
| IB2 | Mugga Ironbark | 5.3 | 0.0 | >30 | >30 | >30 | >30 | 15 | 25 | >30 | >30 |
| PB1 | Paperbark | 8.4 | 0.0 | 15 | >30 | 30 | >30 | 15 | 25 | 25 | >30 |
| PB2 | Paperbark | 9.6 | 0.0 | 10 | >30 | 30 | >30 | 15 | 25 | 10 | >30 |
| SB1 | Red Stringybark | 0.0 | 0.0 | 15 | >30 | 30 | >30 | 20 | 25 | 30 | >30 |
| SB2 | Red Stringybark | 6.5 | 0.0 | 20 | >30 | >30 | >30 | 15 | 25 | >30 | >30 |
| SJ1 | Salvation Jane | 0.0 | 0.0 | 15 | >30 | >30 | >30 | 15 | 25 | >30 | >30 |
| SJ2 | Salvation Jane | 0.0 | 0.0 | >30 | >30 | >30 | >30 | 25 | 25 | >30 | >30 |
| TT1 | Tea Tree | 0.0 | 0.0 | 10 | >30 | >30 | >30 | 15 | 25 | 20 | >30 |
| TT2 | Tea Tree | 0.0 | 0.0 | 20 | >30 | >30 | >30 | 20 | 25 | >30 | >30 |
| YB1 | Yellow Box | 0.0 | 0.0 | >30 | >30 | >30 | >30 | 25 | 25 | >30 | >30 |
| YB2 | Yellow Box | 7.1 | 0.0 | 25 | >30 | >30 | >30 | 20 | 25 | 10 | >30 |
| MX1 | Mixed | 8.0 | 0.0 | 10 | >30 | 25 | >30 | 15 | 25 | 20 | >30 |
| MX2 | Mixed | 5.4 | 5.4 | 20 | 20 | 30 | 30 | 20 | 25 | 25 | 25 |
| *Leptospermum Honeys* | | | | | | | | | | | |
| CL1 | Crows Ash/ Leptospermum | 7.3 | 7.4 | 10 | 10 | 15 | 15 | 20 | 25 | 30 | 30 |
| CL2 | Crows Ash/ Leptospermum | 0 | 0 | 15 | 15 | 20 | 25 | 15 | 25 | 25 | 25 |
| JB1 | Jelly Bush | 0 | 0 | >30 | >30 | >30 | >30 | 25 | 25 | >30 | >30 |
| JB2 | Jelly Bush | 20.3 | 16.6 | 5 | 5 | 10 | 10 | 15 | 20 | 20 | 20 |
| LE1 | Leptospermum | 19.4 | 16.7 | 5 | 5 | 10 | 10 | 15 | 20 | 25 | 25 |
| LE2 | Leptospermum | 6.5 | 6.8 | 10 | 10 | 20 | 20 | 20 | 25 | >30 | >30 |
| *Control Honeys* | | | | | | | | | | | |
| ART | Artificial | 0 | 0 | >30 | >30 | >30 | >30 | 25 | 25 | >30 | >30 |
| JAR | Jarrah | 11.6 | 0 | 10 | >30 | 20 | >30 | 15 | 25 | 10 | >30 |
| MAN | Manuka | 19.7 | 20.1 | 5 | 5 | 10 | 10 | 15 | 15 | 20 | 25 |

[1] PE 2022 = Phenol Equivalence 2022, MIC = Minimum Inhibitory Concentration, TA = Total Activity, NPA = Non-Peroxide Activity

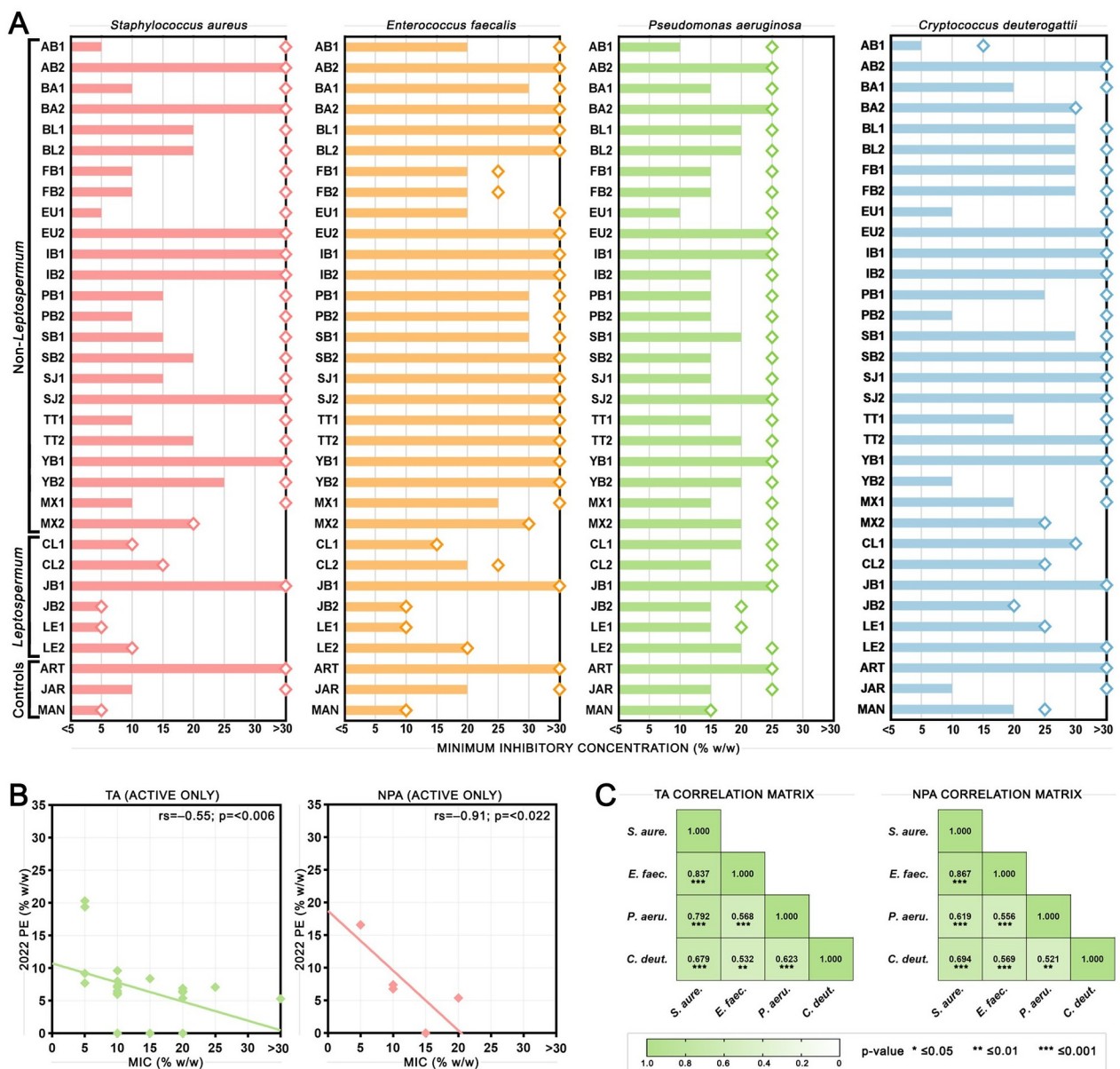

**Fig 2. Antimicrobial activity of honey samples measured by broth microdilution assay, and correlations with the phenol equivalence (PE) assay.** (A) Minimum inhibitory concentrations (MICs) of honey samples against bacterial pathogens *S. aureus*, *E. faecalis*, and *P. aeruginosa*, and fungal pathogen *C. deuterogattii*. Total activity (TA) is shown as bars, and non-peroxide activity (NPA) shown as diamonds. The % (w/w) refers to the concentration of honey diluted in sterile water (for TA) or catalase solution (for NPA). (B) Correlations between PE and MIC measures of TA (left) and NPA (right) in active honey samples. Lines of best fit, Spearman's rho, and p-values are shown. (C) TA (left) and NPA (right) Spearman correlation matrices showing how closely antimicrobial activity of honey samples aligns between different species. Spearman's rho values are shown in the boxes with coloured scalebar also indicating level of correlation. * = p≤0.05, ** = p≤0.01, *** = p≤0.001.

but significantly correlated (Spearman's rho = -0.55, p = <0.006) indicating that honeys with greater TA by phenol equivalence are generally also more active by MIC though the fit is not strong. Low activity honeys (5–10 PE) by phenol equivalence had a much larger spread of MICs (5 –>30% (w/w)), and some honey samples with no activity by phenol equivalence had activity by MIC, suggesting that the MIC tests may be detecting sensitivity to certain

compounds present in some honey samples that is not picked up by the phenol equivalence test. Considering NPA, phenol equivalence and MIC values were much more strongly correlated (Spearman's rho = -0.91, p = <0.022). Comparing the MIC results across species, Spearman's correlation matrices show that both TA and NPA were significantly positively correlated between all four species tested (p≤0.01) (Fig 2C).

## Honey samples with non-peroxide activity contain lower levels of microbes after storage

Honey samples were spread onto agar plates to see whether microbes would be present after storage for 15–17 years, and whether this would align with antimicrobial activity. Nutrient agar plates were chosen to allow growth of a wide range of microorganisms, while V8 juice was chosen to provide extra nutrients for the growth of moulds. Fig 3 shows images of the spread plates after 5 days of growth at 35 ˚C, with plate pairs outlined in colours corresponding to the antimicrobial activity of the honey sample as determined by MIC for *S. aureus*. Minimal growth was noted on spread plates from honeys with NPA. This is particularly distinct when comparing the active *Leptospermum* honeys (CL1, CL2, JB2, LE1, and LE2) to JB1, the one *Leptospermum* honey with no activity, which had high levels of growth. While all inactive honeys had medium to high levels of microbial growth, results were more varied for honeys where the activity was dominated by peroxide-based activity, with some having minimal growth (e.g. BL1 and FB1) and some having high levels of growth (e.g. PB2 and SB1) that did not appear to correspond to activity levels. Organisms growing on all plates were mostly a mix of bacteria and yeasts, with only a single mould noted on the PB1 V8 plate, however, incubation of plates at a lower temperature (25 ˚C) may have resulted in more mould growth.

## Hydrogen peroxide production is highly variable across samples and does not align with floral source

Hydrogen peroxide ($H_2O_2$) production was measured over 6 hours using the horseradish peroxidase (HRP) assay (Fig 4A). All honeys except one exhibited a peak in $H_2O_2$ production and return to near zero by the end of the 6 hours. The one exception was SJ1, a salvation Jane honey, which steadily increased in $H_2O_2$ production throughout the course of the assay. The peak $H_2O_2$ concentration ranged from 30 to 86 µM in inactive non-*Leptospermum* honeys, from 72 to 920 µM in active non-*Leptospermum* honeys, and from 70 to 250 µM in *Leptospermum* honeys. Few honey samples from the same floral source produced similar $H_2O_2$ curves (for example, blueweed/lucerne honeys BL1 and BL2, crow's ash/*Leptospermum* honeys CL1 and CL2, and *Leptospermum* honeys LE1 and LE2), with the majority producing markedly different curves (for example, banksia honeys BA1 and BA2, mugga ironbark honeys IB1 and IB2, and yellow box honeys YB1 and YB2) suggesting that factors other than floral source are a major determinant of hydrogen peroxide production or that the major floral source identified by beekeepers may not necessarily reflect the true complexity of samples. Three metrics were calculated from the $H_2O_2$ time course data: $H_2O_2$ concentration at 2 hours, maximum $H_2O_2$ concentration reached, and area under the curve for total $H_2O_2$ produced (S1 Table). $H_2O_2$ concentration at 2 hours was found to strongly correlate with both maximum $H_2O_2$ concentration reached (Spearman's rho = 0.94, p<0.001) and area under the curve for total $H_2O_2$ produced (Spearman's rho = 0.91, p<0.001), and was therefore chosen to use for further tests as the simplest measurement to assess. $H_2O_2$ concentration at 2 hours was subsequently measured with two additional methods: a 10-acetyl-3,7-dihydroxy-phenoxazine (ADHP) assay which uses the same principle as the commonly used

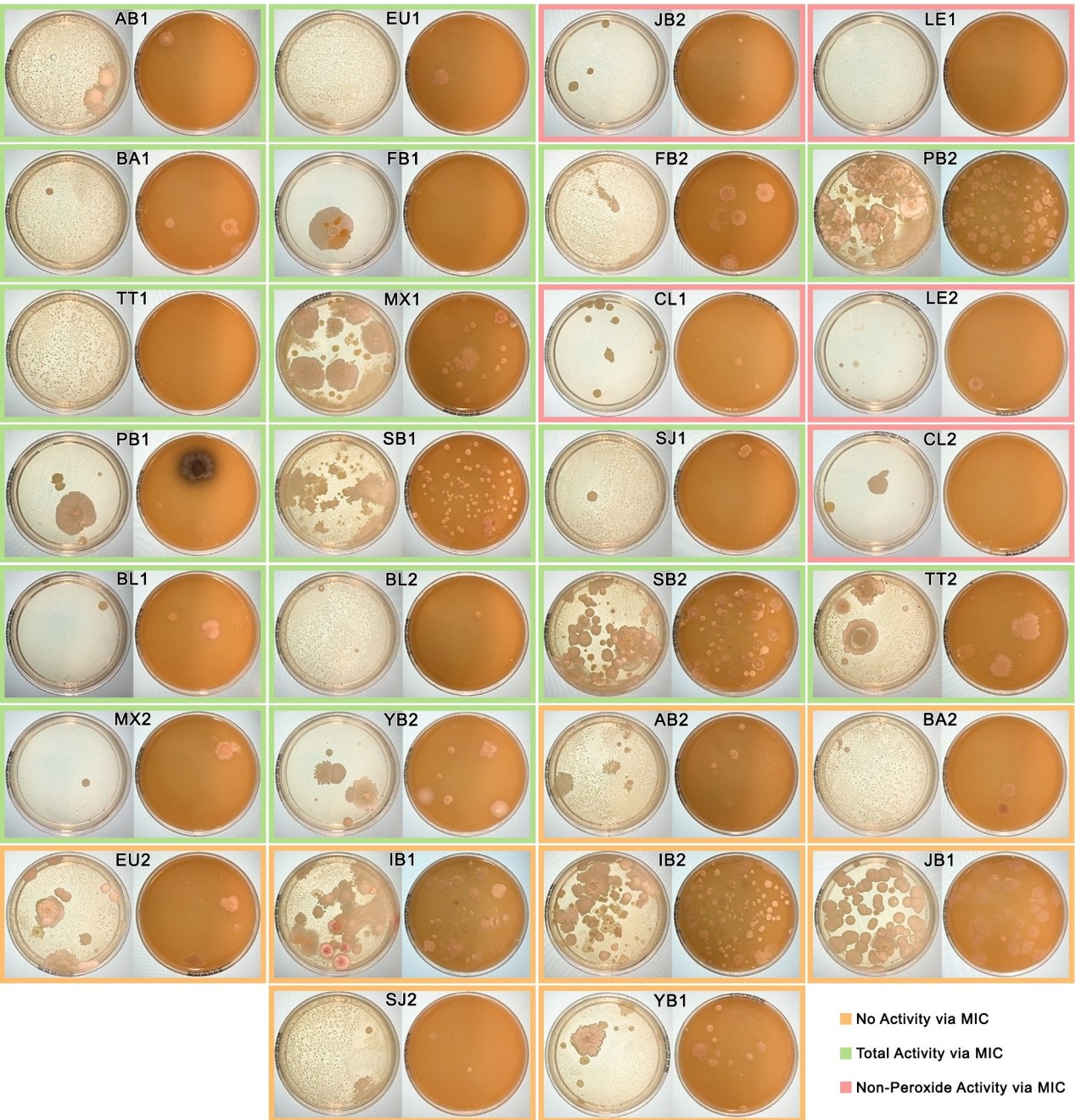

**Fig 3. Growth of microbes from honey samples kept in storage for 15–17 years.** Spread plates of undiluted honey on nutrient (left) or V8-juice (right) agar incubated at 35 ˚C for 5 days. Samples are ordered from most active starting at the top left to least active at the bottom right as determined by *S. aureus* MIC, with coloured borders indicating whether honey samples had no activity (orange), peroxide-dominated activity (green), or non-peroxide activity (red).

commercial AmplexRed kit, and $H_2O_2$ test strips (S1 Table). Spearman correlations between $H_2O_2$ concentration at 2 hours measured by the HRP assay, ADHP assay, and $H_2O_2$ strips are presented in Fig 4B, with all showing strong positive (Spearman's rho $\geq 0.82$) and significant (p$\leq$0.001) correlations.

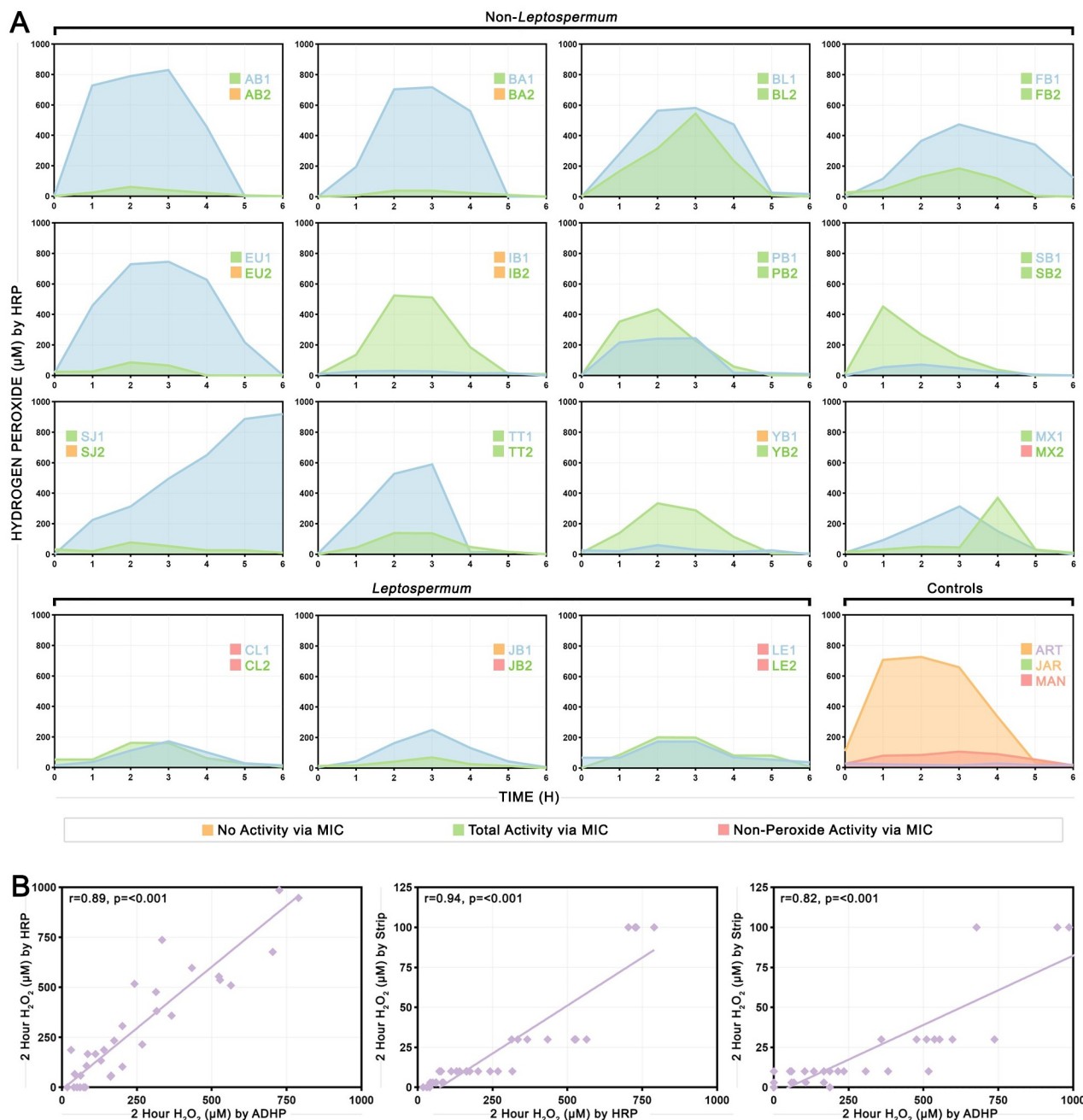

**Fig 4. Hydrogen peroxide (H₂O₂) in honey samples measured by various methods and correlations between measurements.** (A) H₂O₂ production over 6 hours measured by the horseradish peroxidase (HRP) assay. Each box shows the H₂O₂ production curves of two honey samples with the same floral source, except the last box which shows the three control honeys. Coloured squares indicate whether honey samples had no activity (orange), peroxide-dominated activity (green), or non-peroxide activity (red) shown. (B) Correlations between H₂O₂ concentration measured at 2 hours by the HRP and 10-acetyl-3,7-dihydroxyphenoxazine (ADHP) assays (left), the HRP assay and H₂O₂ strips (middle), and the ADHP assay and H₂O₂ strips (right). Lines of best fit, Spearman's rho, and p-values are shown.

## Methods for assaying phenolics and antioxidant content are strongly aligned

Various other physical and chemical properties that have been linked to antimicrobial activity were assayed in honey samples. Phenolics content was measured using the Folin-Ciocalteu (FC) and Fast Blue BB (FBBB) assays (Fig 5A), with the correlation of both methods shown in Fig 5B. Phenolics content ranged from 90–494 mg GAE/kg via FC and from 28–399 mg GAE/kg via FBBB. In both cases, the lowest phenolics content was seen in blueweed/lucerne honey BL1, and the highest was in *Leptospermum* honey LE1. The results of the two assays were significantly positively correlated (Spearman's rho = 0.81, p<0.001), with the FC assay producing uniformly higher values than the FBBB assay, indicating possible interference in the latter by non-phenolic reducing compounds. Antioxidant content was measured using the 2,2-diphenyl-1-picrylhydrazyl (DPPH) and ferric-reducing antioxidant power (FRAP) assays (Fig 5C). Antioxidant content ranged from 1685–4672 µmol TE/kg via DPPH and from 654–5466 µmol $Fe^{2+}$/kg via FRAP. In both cases, the honey with the lowest antioxidant content was Banksia honey BA2, and the honey with the highest antioxidant content was *Leptospermum* honey LE1. The results of the two assays were significantly positively correlated (Fig 5D, Spearman's rho = 0.87, p<0.001). Both phenolics and antioxidant content tended to be largely similar for honey pairs from the same floral source.

## Measurements of honey colour and other physical and chemical properties

The colour of honey samples was assessed visually using a Pfund colour grader chart and compared to three spectrophotometric methods of colour assessment: the Pfund method developed by White [24], mAU, and CIELAB coordinates. Images showing the spectrum of colours seen in honey samples are shown in Fig 5E, with data for each measurement method presented in S1 Table. Honey samples from the same floral source again tended to have similar colours although there were notable exceptions; for example, tea tree honeys TT1 and TT2, and crow's ash/*Leptospermum* honeys CL1 and CL2). By visual assessment, honey samples ranged from 2 mm or Water White (YB2) to 96 mm or Amber (LE1). By the Pfund White method, honey samples ranged from 44 or Extra Light Amber (BA1) to 328 or Dark Amber (LE1). By mAU, samples ranged from 451 (BA1) to 3444 (LE2). By CIELAB, samples ranged from 26 (SJ1) to 56 (CL2). Spearman correlations between colour assessed visually and the three spectrophotometric measurements are shown in Fig 5F. Pfund assessed visually had poor but significant correlation with the Pfund White (Spearman's rho = 0.46, p = 0.007) and CIELAB (Spearman's rho = 0.48, p = 0.005) methods, and a stronger significant correlation with the mAU method (Spearman's rho = 0.74, p<0.001).

Other physical and chemical properties measured are presented in Table 2. pH ranged from 3.46 (SJ1) to 4.12 (BA1). Sugar content ranged from 72.8 (LE2) to 79.0 ˚ Brix (SB2). Water activity ranged from 0.56 (CL2) to 0.69 (CL1). In *Leptospermum* honeys, methylglyoxal (MGO) content ranged from 4 (CL2) to 805 mg/ kg (JB2) and dihydroxyacetone (DHA) content ranged from 2 (CL2) to 1386 mg/ kg (JB2). The significant levels of DHA remaining in *Leptospermum* honeys shows that the conversion to MGO is incomplete, suggesting that this chemical transformation was not occurring or occurring very slowly while the honeys were in storage at 4 ˚C. Several non-*Leptospermum* honeys contained MGO and DHA, including MX2 (101 and 276 mg/ kg, respectively) which exhibited NPA, and BA2, EU2, PB2, TT1, and TT2 (≤81 and ≤129 mg/ kg, respectively) which did not exhibit NPA. This suggests that these honeys, although they may have been predominantly from one floral source, were likely also contributed to by *Leptospermum* nectar. Hydroxymethylfurfural (HMF) content, an indicator of

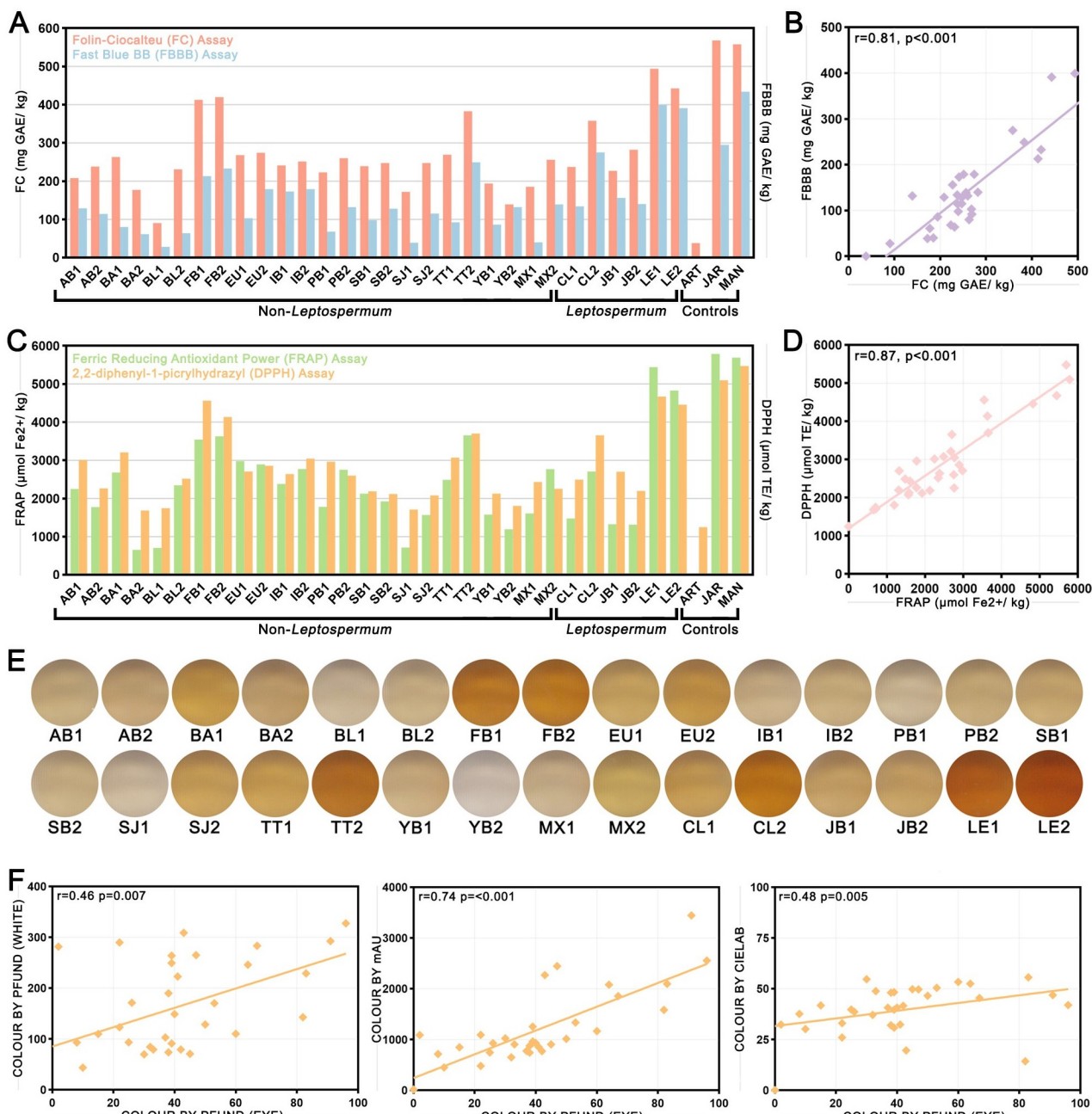

**Fig 5. Phenolics content, antioxidant content, and colour of honey samples measured by various methods and correlations between measurements.** (A) Phenolics content measured by the Folin-Ciocalteu (FC) and fast blue BB (FBBB) assays and (B) the correlation between the two assays. (C) Antioxidant content measured by the 2,2-diphenyl-1-picrylhydrazyl (DPPH) and ferric-reducing antioxidant power (FRAP) assays and (D) the correlation between the two assays. (E) Images showing the spectrum of colours seen in honey samples. (F) Correlations between colour as assessed visually using a Pfund colour grader chart and colour assessed spectrophotometrically using the Pfund method of White (left), mAU (middle), and CIELAB (right). For all correlation charts, lines of best fit, Spearman's rho, and p-values are shown.

**Table 2. Physical and chemical properties of honey samples.**

| ID | ˚ Brix | Water Activity (aW) | pH | MGO (mg/ kg) | DHA (mg/ kg) | HMF (mg/ kg) |
|---|---|---|---|---|---|---|
| *Non-Leptospermum Honeys* | | | | | | |
| AB1 | 78.8 | 0.58 | 3.66 | 0 | 0 | 12 |
| AB2 | 75.0 | 0.67 | 3.55 | 0 | 0 | 6 |
| BA1 | 75.9 | 0.63 | 4.12 | 0 | 0 | 5 |
| BA2 | 76.4 | 0.62 | 3.59 | 1 | 25 | 35 |
| BL1 | 74.8 | 0.66 | 3.56 | 0 | 0 | 3 |
| BL2 | 76.9 | 0.59 | 3.65 | 0 | 0 | 13 |
| FB1 | 78.3 | 0.58 | 3.72 | 0 | 0 | 5 |
| FB2 | 78.3 | 0.59 | 3.80 | 0 | 0 | 11 |
| EU1 | 77.4 | 0.63 | 3.84 | 0 | 0 | 8 |
| EU2 | 75.6 | 0.63 | 3.94 | 14 | 42 | 18 |
| IB1 | 75.7 | 0.65 | 3.75 | 0 | 0 | 3 |
| IB2 | 76.3 | 0.66 | 3.57 | 0 | 0 | 7 |
| PB1 | 76.8 | 0.64 | 3.72 | 0 | 0 | 4 |
| PB2 | 78.5 | 0.61 | 3.66 | 81 | 102 | 4 |
| SB1 | 78.7 | 0.61 | 3.72 | 0 | 0 | 11 |
| SB2 | 79.0 | 0.61 | 3.63 | 0 | 0 | 19 |
| SJ1 | 78.3 | 0.62 | 3.46 | 0 | 0 | 14 |
| SJ2 | 76.9 | 0.66 | 3.61 | 0 | 0 | 41 |
| TT1 | 73.8 | 0.65 | 4.05 | 24 | 31 | 4 |
| TT2 | 74.8 | 0.68 | 3.97 | 68 | 129 | 7 |
| YB1 | 78.3 | 0.61 | 3.74 | 0 | 0 | 9 |
| YB2 | 78.6 | 0.61 | 3.71 | 0 | 0 | 3 |
| MX1 | 75.9 | 0.60 | 3.74 | 0 | 0 | 3 |
| MX2 | 77.8 | 0.61 | 3.86 | 101 | 276 | 5 |
| *Leptospermum Honeys* | | | | | | |
| CL1 | 73.4 | 0.69 | 3.65 | 167 | 136 | 23 |
| CL2 | 78.8 | 0.56 | 3.73 | 4 | 2 | 24 |
| JB1 | 73.9 | 0.68 | 3.67 | 191 | 232 | 12 |
| JB2 | 76.9 | 0.64 | 3.66 | 805 | 1386 | 14 |
| LE1 | 74.3 | 0.65 | 3.56 | 479 | 240 | 15 |
| LE2 | 72.8 | 0.69 | 3.62 | 250 | 190 | 4 |
| *Control Honeys* | | | | | | |
| ART | 78.1 | 0.58 | 4.23 | 0 | 0 | 2 |
| JAR | 83.1 | 0.54 | 4.08 | 0 | 0 | 28 |
| MAN | 79.4 | 0.60 | 3.82 | 657 | 280 | 68 |

heat and storage changes, ranged from 3 (BL1, IB1, YB2, MX1) to 41 mg/ kg (SJ2) with an average of 11 mg/ kg.

## The elemental content of honey samples is diverse and variable

Honey samples were assessed for their level of various common elements. A heatmap using standardised values is shown in Fig 6A, with raw values presented in S1 Table. Overall, potassium (861 μg/g) was the most prevalent element on average followed by sodium (148 ug/g), phosphorus (39 μg/g) and magnesium (38 μg/g). Elements present at lower average quantities were calcium (8.5 μg/g), zinc (8.3 μg/g), iron (4.4 μg/g), and manganese (3.4 μg/g). Elements

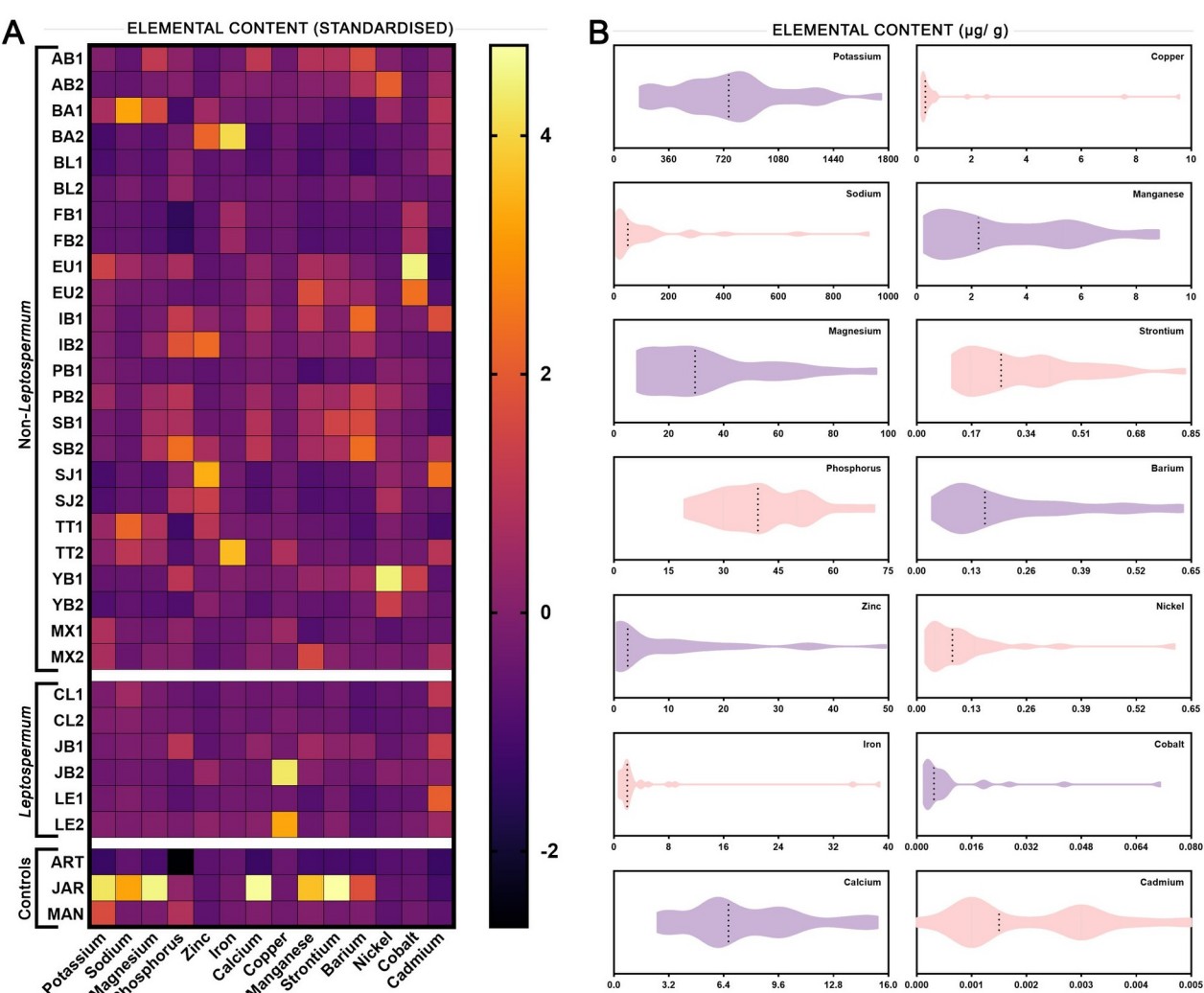

**Fig 6. Elemental content of honey samples.** (A) Heatmap showing the relative elemental content of honey samples using normalised measurements. (B) Box plots showing the distribution of values for each measured element in honey samples. Dotted lines indicate the median value.

present at very low average quantities were copper (0.936 µg/g), strontium (0.358 µg/g), barium (0.228 µg/g), nickel (0.106 µg/g), cobalt (0.009 µg/g), and cadmium (0.002 µg/g). Fig 6B shows the distribution of values for each measured element in honey samples. Some such as potassium and calcium show an even distribution of values, while others such as copper and iron are present in most honeys at very low levels with only a few samples having large amounts. This indicates that those elements might be more prevalent in the environment in certain areas. Large quantities of a particular element in only one honey from a floral source pair (for example, copper in JB1 and LE2 but not JB2 or LE1) indicate that these elements are likely not naturally present in higher quantities in these particular plants but originate from other environmental sources. The large quantities of several elements in commercial Jarrah honey suggests that these may be more prevalent in the Western Australian environment where Jarrah grows.

## Correlations between the antimicrobial activity of non-*Leptospermum* honey and its physical and chemical properties

A Spearman correlation matrix was generated to assess the strength and direction of associations between antimicrobial activity data and physical and chemical property data, separated by TA and NPA (Fig 7A). A partial least squares (PLS) regression biplot was additionally generated for non-*Leptospermum* honeys using antimicrobial activity data as the outcome variables and physical and chemical property data as the explanatory variables in order to visualise the relationship between honey activity and other measured properties (Fig 7B). Variables towards the outer edges of the plot indicate a strong contribution to the model, while variables towards the centre indicate a weak contribution. Variables close together are positively correlated to each other and negatively correlated to those far away and opposite them. MIC values have been reversed for both analyses so that positive correlations indicate properties that align with increased antimicrobial activity. Average TA and Average NPA (S1 Table) were calculated by taking the average of the reversed MIC values for *S. aureus*, *E. faecalis*, *P. aeruginosa*, and *C. deuterogattii* for each honey sample. Correlations between MIC values for each individual species and average TA or NPA MIC values found them to be in strong alignment and thus the average values were considered a good representation of the overall antimicrobial activity of honey samples (data not shown).

All measures of $H_2O_2$ were significantly positively correlated with TA measured via the phenol equivalence assay ($p \leq 0.05$), and TA measured via the MIC assay against *S. aureus*, *P. aeruginosa*, and *C. deuterogattii* ($p \leq 0.05$), but not *E. faecalis*, suggesting this organism is less susceptible to $H_2O_2$. This is further seen on the biplot with *S. aureus*, *P. aeruginosa*, and *C. deuterogattii* TA clustering close to measures of $H_2O_2$ production while *E. faecalis* TA is further away. Other physical and chemical variables significantly correlated with antimicrobial activity against at least two organisms via MIC were phenolics via FC and antioxidants via DPPH, which were positively correlated ($p \leq 0.05$), and water activity, which was negatively correlated ($p \leq 0.05$). No variables other than $H_2O_2$ were significantly correlated with phenol equivalence TA, indicating that it is less sensitive than the MIC assay at picking up the antimicrobial contributions of other components measured in this study. For average MIC TA, all measures of $H_2O_2$ ($p \leq 0.05$), as well as antioxidants via DPPH ($p \leq 0.05$) were significantly correlated.

Considering NPA, MGO and DHA were significantly positively correlated ($p \leq 0.01$) for all organisms using phenol equivalence and MIC assessments. Other physical and chemical variables significantly correlated with antimicrobial activity against at least two organisms assessed via MIC were phenolics assessed via FC and FBBB ($p \leq 0.05$) and colour assessed via Pfund Eye, mAU and CIELAB ($p \leq 0.05$) which were all positively correlated. Although antioxidants assessed via DDPH was significantly correlated with several measures of TA, it was not significantly correlated with NPA for any organism except *E. faecalis*. On the biplot, *E. faecalis* NPA clusters closely with measures of phenolics, antioxidants, and colour indicating that this organism is more sensitive to phenolic and antioxidant compounds. *C. deuterogattii* NPA appears close to the centre of the plot indicating an overall weak contribution to the model. A significant positive correlation was seen between *C. deuterogattii* NPA and HMF content ($p \leq 0.05$), however whether this is a causal relationship requires further investigation.

Linear and multiple linear regressions were used to determine which physical and chemical properties are predictive for antimicrobial activity assessed using the phenol equivalence assay (*S. aureus* only) or MIC (averaged across the four microorganisms) (Fig 8). Linear regressions use a single independent variable, while multiple linear regression uses two or more independent variables, to predict the outcome of a dependent variable. The actual and predicted dependent variable values can then be compared to determine the goodness of fit and

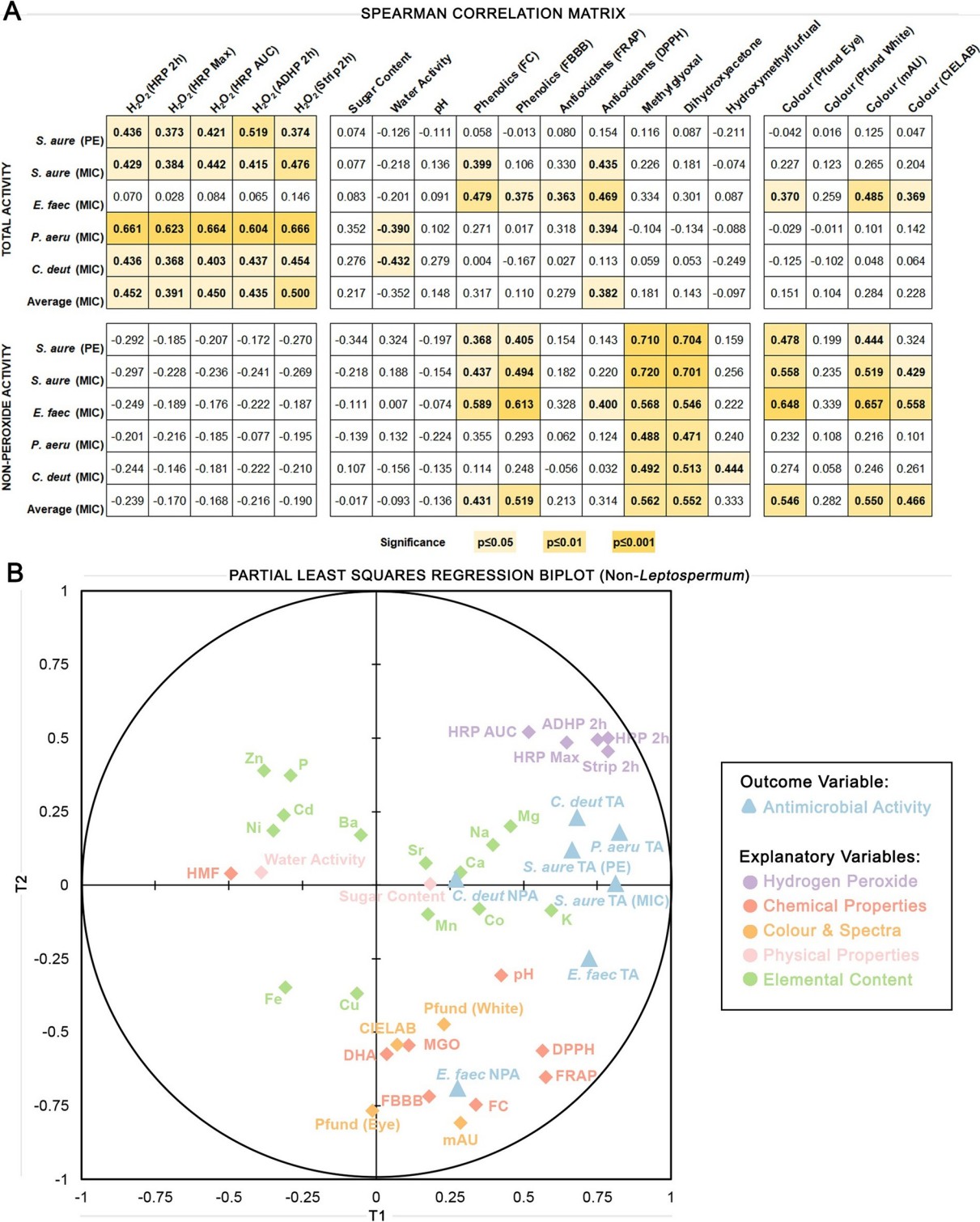

**Fig 7. Correlations between antimicrobial activity and the physical and chemical properties of honey samples.** (A) Spearman correlation matrix showing how closely and in what direction antimicrobial activity (horizontal) aligns with physical and chemical properties (vertical). MIC values have been reversed so that positive correlations indicate properties that align with increased antimicrobial activity. Sperman's rho values are shown in the boxes with 1 indicating the strongest positive correlation and -1 indicating the strongest negative correlation. Statistically significant correlations are shaded in yellow with the corresponding significance level indicated on the legend to the right. (B) Partial least squares (PLS)

regression biplot for non-*Leptospermum* honeys showing the relationships among all variables measured in this study and giving information about dataset structure. PLS regression was performed using antimicrobial activity as the outcome variable, and the physical and chemical properties of honey samples as the explanatory variables. Variables towards the outer edge of the plot indicate a strong contribution to the model, while variables towards the centre indicate a weak contribution. Variables close together are positively correlated to each other and negatively correlated to those further away and in opposite quadrants.

significance of the model. These models necessitate a linear relationship and are therefore more stringent than Spearman correlations but provide a simpler and more predictive relationship. For *Leptospermum* honeys, MGO content correlated well with TA assessed by phenol equivalence ($R^2 = 0.81$; p = 0.014) as expected, but not with average TA assessed by MIC ($R^2 = 0.32$; p = ns).

For non-*Leptospermum* honeys, $H_2O_2$ production correlated poorly with TA assessed by phenol equivalence and was not significant ($R^2 = 0.13$; p = ns), while the correlation with average TA assessed by MIC was still poor but reached significance ($R^2 = 0.25$; p = 0.016). Considering multiple factors for each individual species yielded much improved correlations with the best combination of factors being different for each organism. For *S. aureus*, the best model used $H_2O_2$, phenolics (FC), and antioxidants (DPPH) ($R^2 = 0.47$; p = 0.006), while for *E. faecalis* it used $H_2O_2$, phenolics (FBBB), and antioxidants (DPPH) ($R^2 = 0.41$; p = 0.005) and for *P. aeruginosa* it used $H_2O_2$, water activity, and antioxidants (DPPH) ($R^2 = 0.60$; p = <0.001). While $H_2O_2$ itself was not directly correlated with antimicrobial activity against *E. faecalis*, it's

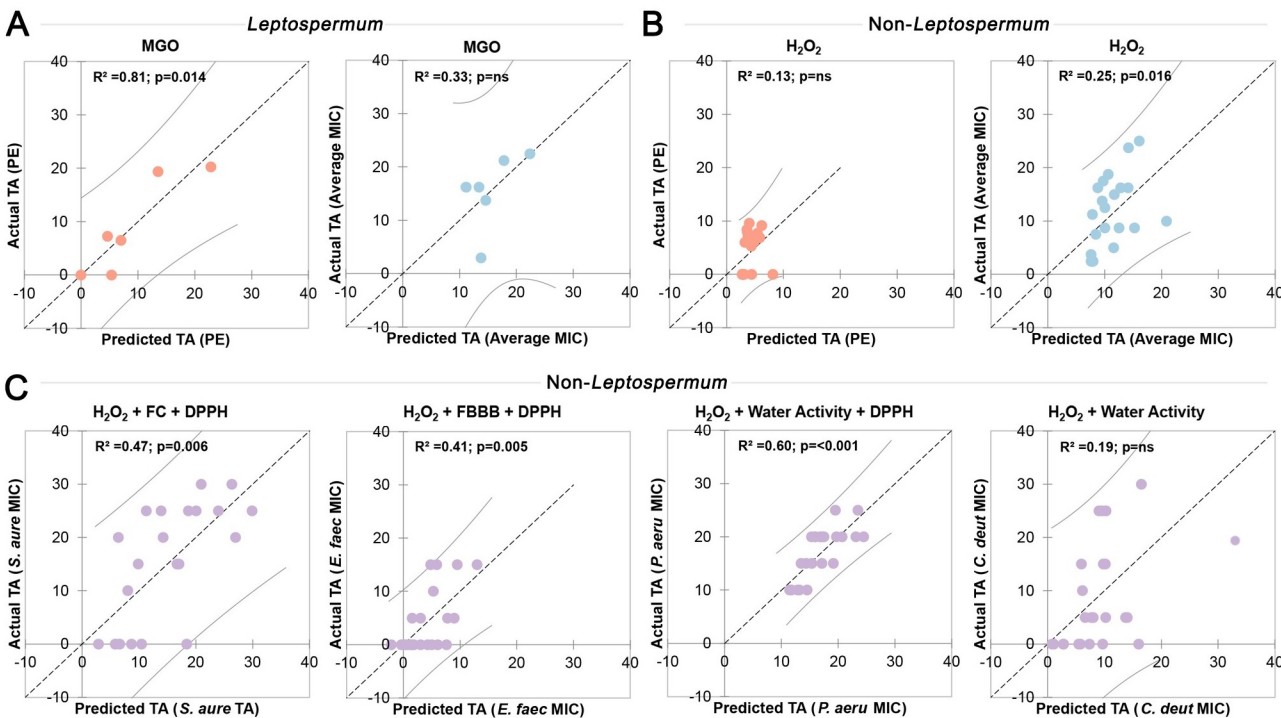

**Fig 8. Linear regressions showing physical and chemical properties that have predictive power for antimicrobial activity.** (A) For *Leptospermum* honeys, methylglyoxal (MGO) correlates well with total activity assessed by phenol equivalence assay (PE), but not with MIC averaged across the 4 species tested (Average MIC). (B) For *non-Leptospermum* honeys, total $H_2O_2$ production (area under curve measured by HRP assay) correlates poorly with TA assessed by phenol equivalence or average MIC. (C) Including multiple variables in the regressions substantially improves correlations for individual species. Models that are significantly predictive of antimicrobial activity were able to be identified for the bacterial species *S. aureus*, *E. faecalis*, and *P. aeruginosa*, but not for the fungal species *C. deuterogattii*.

inclusion alongside phenolics and antioxidants improved the strength of the multivariable model, suggesting that these components may operate together synergistically. The best model for fungal species *C. deuterogattii* used $H_2O_2$ and water activity but was poor and did not reach significance ($R^2 = 0.19$; p = ns), suggesting that there may be significant contributing antifungal factors that have not been measured in the current study.

## Discussion

This study set out to address key challenges to assessing the medicinal potential of honey that limit development by the honey industry and impede clinical adoption. Our objectives were to investigate the long-term shelf life of honey, to elucidate key factors influencing the antimicrobial activity of honey, and to give the honey industry practical ways to assess these. We profiled the antimicrobial properties of a diverse collection of Australian honeys from the New South Wales region, evaluating the effects of long-term storage on activity and analysing how various physical and chemical traits align with antimicrobial activity against distinct microbes.

### Factors useful for predicting antimicrobial activity

We utilised linear regression models to ascertain the predictive capacity of various chemical and physical properties on antimicrobial activity. For *Leptospermum* honeys, our models reinforce the established use of MGO as a key predictor of antimicrobial activity measured via the phenol equivalence assay ($R^2 = 0.81$) [16]. A poor correlation was seen between MGO and activity measured via the MIC assay, but due to the small sample size of *Leptospermum* honeys it is difficult to tell whether this was due to outliers, or whether the MIC assay is more sensitive to non-MGO based activity. Further analysis with more *Leptospermum* samples is required to determine whether MGO is predictive for activity measured via the MIC assay. Non-*Leptospermum* honeys showcased a more intricate interplay among variables, where combining various factors including $H_2O_2$ production, antioxidant content, and water activity markedly improved predictive capabilities ($R^2 = 0.41$–$0.60$) for antibacterial activity compared to considering $H_2O_2$ alone ($R^2 = 0.25$) but was still unable to reach very high confidence. These findings underscore the multifaceted nature of honey bioactivity, emphasising the need for comprehensive models incorporating diverse chemical constituents. In future studies, exploring potential synergistic interactions among chemical components combined with further validation across diverse honey samples, may enable more nuanced predictive models. However, even if these models become substantially more robust, the high potential for significant deviation in unique samples coupled with the need to accurately measure multiple properties suggests that a direct measurement of antimicrobial activity will remain the superior strategy for industry in terms of both accuracy and simplicity.

### Correlations between antimicrobial activity and specific physical or chemical properties

In our investigation of the relationship between the various physical and chemical properties and antimicrobial activity, we found several significant correlations. $H_2O_2$ production was significantly correlated with TA against *S. aureus*, *P. aeruginosa*, and *C. deuterogattii*. This was expected since $H_2O_2$ is the main contributor to the antimicrobial activity of most non-*Leptospermum* honeys and these microbes have all previously been reported to be susceptible to honey [5, 8]. Activity against *E. faecalis*, however, correlated with increased phenolics content, antioxidant capacity, and honey colour but not with $H_2O_2$ production. Phenolic compounds and antioxidants are known to contribute to the antimicrobial activity of honey, interacting with bacterial membranes and enzymes to disrupt their function and cause cell death [29].

Darker honey colour can indicate an increased presence of phenolic compounds and other phytochemicals [30]. The clustering of measures of honey colour, phenolic and antioxidant compounds, and MGO and DHA in Fig 7B indicates that these parameters are indeed correlated in our samples. *E. faecalis* has a higher level of resistance to oxidative stress than other bacteria [31], due to its ability to produce several detoxifying enzymes that can break down reactive oxygen species. It has also been identified as a pathogen in honey bee hives [32], acting as a secondary invader during outbreaks of European Foulbrood and resistance to $H_2O_2$-mediated activity could explain its ability to opportunistically invade weakened colonies where $H_2O_2$ production provides a significant level of defence. Increased phenolics and/or antioxidant content also correlated with TA and NPA against *S. aureus* and *P. aeruginosa*, suggesting that antioxidant compounds contribute to the effects of honey against these organisms. Antioxidants measured via DPPH correlated significantly with TA for *S. aureus*, *P. aeruginosa*, and *E. faecalis*, and with NPA for *E. faecalis* alone. This suggests that *E. faecalis* is somewhat susceptible to these antioxidant compounds on their own, while the other organisms require additional weakening from $H_2O_2$ for the antioxidant compounds to be effective.

Water activity was found to be significantly correlated with the TA of *P. aeruginosa* and *C. deuterogattii*, with decreased water activity correlating with increased honey activity. Water activity is an important factor that affects the antimicrobial activity of undiluted honey, with a low water activity inhibiting microbial proliferation by making water required for their growth unavailable [33].

However, as honey is diluted to at least 32% (w/w) in the MIC assay it is unlikely that water activity directly contributes to its antimicrobial properties. Instead, it likely serves as an indicator for other factors that may alter antimicrobial activity. Water activity decreases with the increased bee processing, which could result in higher levels of bee enzymes and antimicrobial peptides. Differences in nectar sources can also affect water activity, and these may introduce other antimicrobial features. Remaining physical and chemical properties that were measured, including pH, sugar content, and elemental content were not significantly correlated with activity. Elemental content could theoretically influence antimicrobial properties as certain elements are vital cofactors for enzymes that synthesise phytochemicals and antioxidants [34] and metals such as iron and copper can also enhance $H_2O_2$-based killing via Fenton reactions [35]. However, the diverse actions of these elements and the varied floral sources of the samples in the current study, each with unique phytochemical constituents, likely hindered the identification of specific elemental effects. To gain a more comprehensive understanding, future studies could employ manipulative experiments involving the addition of specific elements to individual honey samples, facilitating a more nuanced exploration of their potential contributions to antimicrobial activity.

### Recommended methods for assaying honey properties

An evaluation of the pros and cons of methods trialled in this study are presented in Table 3. Looking at antimicrobial activity, we found a significant but weak correlation between phenol equivalence and MIC results, similar to previous work that specifically set out to compare the two methods [15], and to a study comparing general agar diffusion and broth dilution methods for assessing honey activity [36]. Although it is the current Australian honey industry standard, the phenol equivalence assay has several significant drawbacks including difficulty interpreting the results, variability between laboratories, and the use of the toxic chemical phenol [14, 15]. In addition, the water-based agar matrix may prevent large or polar molecules from diffusing freely, which could confound interpretation in honey samples where these contribute to activity [13]. This may explain why we saw correlations between phenolics, antioxidants and TA via

**Table 3. Evaluation of methods for assessing honey properties.**

| Method | Accuracy | Difficulty | Safety | Comments |
|---|---|---|---|---|
| **Antimicrobial Activity** | | | | |
| Phenol Equivalence | Fair—but underestimates non-peroxide factors in non-*Leptospermum* honeys | High | Hazardous chemical—phenol; requires PC2 facilities for *S. aureus* | Only performed with the bacterial pathogen *S. aureus* |
| **Minimum Inhibitory Concentration** * | Good | Medium | Safe; requires PC2 facilities for microbial pathogens | Can readily be applied to a wide variety of microbes including bacteria, yeasts, and moulds |
| **Hydrogen Peroxide** | | | | |
| **HRP** * | Good | High | Hazardous chemical– 6M $H_2SO_4$ | Cheap for bulk samples, expensive for few samples |
| **ADHP** * | Good | Medium | Safe | Cheap for bulk samples, expensive for few samples |
| **Strip Test** * | Good | Low | Safe | Cheap for few samples, expensive for bulk samples |
| **Phenolics** | | | | |
| FC | Fair—but affected by non-phenolic reducing compounds | Low | Safe | May have utility for comparing multiple samples, but should not be used to determine absolute values |
| **FBBB** * | Good | Medium | Safe | More accurate than FC for determining absolute values |
| **Antioxidants** | | | | |
| FRAP | Good | Medium | Safe | Produces results in alignment with DPPH |
| **DPPH** * | Good | Medium | Hazardous chemical—DPPH | Produces results in alignment with FRAP but is more strongly associated with antimicrobial activity |
| **Colour by Spectrophotometer** | | | | |
| Pfund | Poor—vulnerable to turbidity and crystallisation | Low | Safe | Very inaccurate, should be avoided |
| **mAU** * | Good | Low | Safe | Gives the best correlation with colour assessed visually |
| CIELAB | Poor—saturated at lower wavelengths | Medium | Safe | May have utility when in-depth comparison across the colour spectrum is desired and honeys can be diluted to avoid saturation |

* Recommended method

MIC, but not via phenol equivalence, as the free diffusion of these compounds through the agar matrix may be hindered. This may also explain why the range of activity was much wider when assessed via MIC than via phenol equivalence, and why NPA was identified in some honeys by MIC but not by phenol equivalence. The MIC assay can be readily applied to a wide variety of microbes including bacteria, yeasts, and moulds, and is the gold standard for antimicrobial susceptibility testing [21, 22]. Although honey activity is strongly correlated across diverse organisms, certain species such as *E. faecalis* and *C. deuterogattii*, exhibit distinct susceptibility and the MIC assay can capture these nuances when in-depth analyses are desired. Overall, we argue that the MIC assay is a superior method for testing the antimicrobial activity of honey and where possible should replace the phenol equivalence assay as the industry standard.

Measuring $H_2O_2$ production in honey using the HRP assay, we found a strong alignment between $H_2O_2$ produced at 2 hours, maximum $H_2O_2$ concentration reached, and total $H_2O_2$ produced. We additionally saw strong correlations between $H_2O_2$ concentration at 2 hours measured using three independent methods. A major difficulty when screening honey samples for $H_2O_2$ is the need to conduct an entire time course assay in order to identify the timepoint where $H_2O_2$ production peaks. Our results suggest that the 2-hour time point would provide a reliable indication of $H_2O_2$ production without the need for a full assay, and this can be measured using either simple strip tests or more specific colourimetric assays, depending on the level of granularity desired. However, a caveat is that there can be outliers, such as honey sample SJ1, where $H_2O_2$ curves can occur over a much longer time frame [20].

Evaluating phenolics content and antioxidant content, we used simple colorimetric assays that provided an easy and rapid approximation of these properties. For phenolics content, our results indicate that the FBBB assay is superior to the FC assay as it is less affected by non-phenolic compounds. While the FC assay is used widely as a measure of total phenolics content due to its simplicity, it is known to be affected by reducing compounds including those present in honey such as sugars and amino acids [37]. Indeed, we saw that results from the FC assay were universally higher than those from FBBB assay, which is based on a direct reaction with active hydroxyl groups in phenolic compounds. The FC assay may have utility for comparing multiple honey samples but should not be used for determining absolute values. For antioxidant content, the DPPH and FRAP assays were in strong alignment with each other though the DPPH assay was more strongly associated with antimicrobial activity, suggesting that it may be picking up the activity of a distinct set of compounds. Therefore, while either assay is suitable for determining the antioxidant activity of honey samples, the DPPH assay may provide slightly more in-depth information.

Finally, when evaluating honey colour, poor correlations were found between Pfund value assessed visually and assessed spectrophotometrically using the White and CIELAB methods. The White method relies on only a single absorbance value taken at 635 nm, making it particularly vulnerable to turbidity caused by the growth of microbes or incompletely dissolved sugar crystals. The CIELAB method uses a wave scan from 280–780 nm, but we found honey absorbance values at the low-end wavelengths were often saturated far past the range of accuracy. Colour intensity measured in mAU, which uses two absorbance values taken at 450 and 720nm, gave a much better correlation with colour assessed visually. The accuracy of this method combined with its simplicity makes it the most efficient method for assessing honey colour via spectrophotometer.

## Changes in the antimicrobial activity of honey over time

The majority of initially active honey samples retained some level of activity after 15+ years of storage at 4 ˚C in the dark, with non-*Leptospermum* honeys experiencing an average of 64% decrease in TA and *Leptospermum* honeys experiencing an average decrease of 31% in NPA. The strong correlations found between the 2005–2007 and 2022 data indicate that TA and NPA declined at a relatively consistent rate across samples. This suggests that, after accounting for activity type, storage conditions are the biggest determinant of activity decline rather than chemical differences in the individual samples.

Relatively few studies have directly compared the antimicrobial activity of honey samples over long periods of time, however a general decline in activity despite storage conditions is consistently noted. Looking at 10 samples of non-*Leptospermum* Australian honey from the same original survey as the current study, Irish et al. [12] reported an average decline of 23% after 8–22 months storage in the dark at 4 ˚C, and a follow-up study on a subset of 15 samples found an average decline of 30% after 11–12 years storage in the dark at 4 ˚C [20]. This decrease in activity over time is thought to be due to a combination of factors including exposure to light, heat, and air, which can all cause chemical changes and a breakdown of active components [9]. Various studies have found phenolics and antioxidant content can decrease from 17–92% following up to one year of storage under various conditions, along with significant changes to pH, free acidity, moisture content, HMF content and honey colour [38–40].

*Leptospermum* honeys are unique in that their active component is methylglyoxal (MGO), which is produced from the conversion of dihydroxyacetone (DHA) naturally found in *Leptospermum* nectar [41]. This chemical conversion takes place over time during the maturation of honey and can therefore lead to an increase in antimicrobial activity over time. This was

previously observed by Irish et al. [12] in *Leptospermum* samples stored at 25 ˚C for 8–22 months, with some increasing in NPA by up to 37% and others decreasing by up to 16%. In the current study, a decrease in activity was observed for most of the *Leptospermum* honeys and we found that several still contained a considerable amount of DHA, likely due to low temperature storage slowing or halting the conversion to MGO. Furthermore, while MGO is thought to be relatively insensitive to light and heat, it can react with oxygen over time to form other compounds [42], which would lead to a decrease in antimicrobial activity, though this would likely only apply to compounds on the surface.

## Conclusions

Our study offers recommendations for the assessment of medicinal honey and provides further evidence of the complex interplay between different properties of honey and its antimicrobial activity against various microorganisms. While MGO is a reliable predictor of activity in *Leptospermum* honeys, the variability inherent in non-*Leptospermum* honeys limits the development of models with predictive capabilities, and a direct assessment of antimicrobial activity is still required to determine medicinal potential. We recommend use of the MIC method to capture the full range of activity and to enable testing across diverse microbial pathogens. Furthermore, our study highlights the remarkable resilience of honey's antimicrobial properties, as evidenced by the retention of activity even after more than 15 years of storage. This longevity underscores the therapeutic potential of honey to offer a natural and sustainable alternative for combatting microbial infections. However, its potential to support microbial growth and degradation highlights the importance of proper storage and handling to maintain its quality and optimal activity.

## Supporting information

**S1 Table. All data collected in this study.**
(XLSX)

## Acknowledgments

The authors would like to thank Daniel Susantio and Bridie Stanfield for technical assistance, Dr Nicholas Proschogo (School of Chemistry, University of Sydney) for valuable advice on sample digestion and for performing the ICP-MS measurements, and the authors of [12] as well as the beekeepers who provided the original samples.

## Author Contributions

**Conceptualization:** Kenya E. Fernandes.

**Data curation:** Kenya E. Fernandes.

**Formal analysis:** Kenya E. Fernandes.

**Funding acquisition:** Nural N. Cokcetin, Dee A. Carter.

**Investigation:** Kenya E. Fernandes.

**Methodology:** Kenya E. Fernandes, Andrew Z. Dong, Aviva Levina, Nural N. Cokcetin, Peter Brooks.

**Project administration:** Nural N. Cokcetin.

**Resources:** Dee A. Carter.

**Supervision:** Dee A. Carter.

**Visualization:** Kenya E. Fernandes.

**Writing – original draft:** Kenya E. Fernandes.

**Writing – review & editing:** Kenya E. Fernandes, Andrew Z. Dong, Aviva Levina, Nural N. Cokcetin, Peter Brooks, Dee A. Carter.

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
