## [Decision Letter · Decision Letter 0]

20 Mar 2024

PONE-D-24-05003Long-term stability and the physical and chemical factors predictive for antimicrobial activity in Australian honeyPLOS ONE

Dear Dr. Fernandes,

Thank you for submitting your manuscript to PLOS ONE. After careful consideration, we feel that it has merit but does not fully meet PLOS ONE’s publication criteria as it currently stands. Therefore, we invite you to submit a revised version of the manuscript that addresses the points raised during the review process.

We look forward to receiving your revised manuscript.

Kind regards,

Branislav T. Šiler, Ph.D.

Academic Editor

PLOS ONE

Journal Requirements:

Additional Editor Comments:

The authors are advised to put more effort on vindicating the selection of 15-17 years old honey in the experiments, as required by Reviewer #1. Additionally, the aim of the study in the last paragraph of Introduction should be reinforced. Please note that Reviewer #2 provided their comments in the attached file. Terms in subsections of M&M stand capitalized, while deviation from this rule is seen in Results as well as in Discussion. Please uniform this.

Reviewers' comments:

Reviewer's Responses to Questions

**Comments to the Author**

1. Is the manuscript technically sound, and do the data support the conclusions?

Reviewer #1: Partly

Reviewer #2: Yes

2. Has the statistical analysis been performed appropriately and rigorously? 

Reviewer #1: Yes

Reviewer #2: Yes

3. Have the authors made all data underlying the findings in their manuscript fully available?

Reviewer #1: Yes

Reviewer #2: Yes

4. Is the manuscript presented in an intelligible fashion and written in standard English?

Reviewer #1: Yes

Reviewer #2: Yes

5. Review Comments to the Author

Reviewer #1: Manuscript PONE-D-24-05003 describes mainly the antimicrobial properties of honey samples after they has been stored for 15-17 years. I must say that the experimental section is of very high quality, both in conceptualisation as well as execution. Interpretation of the results is also very thorough, even though some points need attention, as mentioned below.

However, the major concern in this manuscript is the scope of the whole work, as well as the importance of the conclusions.

At first, the selection of the samples is surprising. Why would anyone choose 15–17-year-old honey samples to conduct so many experiments? What would they want to show? And how this can help industry meet the growing demand for high-quality, medicinally active honey? Does industry market honey that old?

In addition, why would anyone be interested in investigating how the antimicrobial activity of honey changes after 15-17 years of storage? Would anyone ever store honey for so long? Also, did you evaluate organoleptic properties of those samples to see how they have changed through time?

I understand that scientific curiosity could easily drive someone to do such a research, yet this is far beyond the scope that the authors state or the conclusions they wanted to achieve. It is not mentioned in the Abstract that samples were 15-17-year-old! You conducted an excellent series of experiments and made conclusions with very old honey samples. Why?

From where I see it, the authors must either reconsider the scope of this manuscript or provide strong arguments for their choice to employ so old samples in their work.

Additional major comments

Title

The title does not reflect the content of the manuscript. Is this work about long-term stability and how this is developed in the manuscript? Also, I agree that MIC is predictive for antimicrobial activity, but I believe it can not be considered a chemical factor.

Lines 311-312.

The authors here state a major problem in the interpretation of their results. Based on the publication of Irish et al. (2011), honey origin was verified by the apiarists that collect them. This raises a concern regarding the correct assignment of the floral source of the samples. In any case, you should add in Materials and methods how the botanical origin of samples was verified.

Lines 395-412.

This section needs more analysis and careful wording. It is evident that elemental analysis of honeys can differentiate samples for their geographical origin as metallic components are correlated with the soil where the plants grow. This is in agreement with the finding that Jarrah honey contains large quantities of several elements.

Additionally, it is well known that honeydew honeys contain large quantities of ash compared to flower honeys. I am sorry, but I am not familiar with the Australian flora. Are there any honeydew honeys in this research? This information would be interesting for the reader and could help you better explain your results.

The comment that maybe these large quantities of elements could have been introduced during processing and packaging is arbitrary and must either be removed or supported by literature. I agree that this can happen, however few of the elements found in honey can result through this route.

Lines 560-569.

These lines seem problematic. Typical water activity of honey is 0.5 to 0.65, while most microorganisms need more than 0.7 to grow. In your work, you measured water activity was as high as 0.69. It is not honey with low water activity that do not favour microbial growth, it is rather all honeys that do so! In addition, the shelf stability of honey is not correlated with microbial inhibition! It is HMF and diastase activity that determine shelf life of honey. Microbes almost never develop in honey, with the exception of osmophilic yeasts under favorable conditions. Moreover, can you determine what you mean with “low water activity” in line 565? In my opinion, shelf stability of honey cannot be correlated with water activity.

Minor comments

Line 224. You wrote 2002 instead of 2022.

Lines 389-390 and elsewhere. HMF is measured in mg/Kg of honey and ppm is hardly ever used. Even though it is not wrong, someone would be surprised to see it (as I did).

Lines 646-653. Include the conditions of storage of the referenced works that you include here.

Line 662. How can any compound in honey react with oxygen apart from those on the surface?

Line 667. Predictor, not predicator.

Reviewer #2: Please see my corrections and comments in the corrected manuscript.

6. PLOS authors have the option to publish the peer review history of their article (what does this mean?). If published, this will include your full peer review and any attached files.

Reviewer #1: **Yes: **Eleftherios Alissandrakis

Reviewer #2: No

---

## [Author Response · Author response to Decision Letter 0]

14 Apr 2024

Please see our comments in the attached 'Response to Reviewers' document.

---

## [Editor Report · Decision Letter 1]

16 Apr 2024

PONE-D-24-05003R1Long-term stability and the physical and chemical factors predictive for antimicrobial activity in Australian honeyPLOS ONE

Dear Dr. Fernandes,

Thank you for submitting your manuscript to PLOS ONE. After careful consideration, we feel that it has merit but does not fully meet PLOS ONE’s publication criteria as it currently stands. Therefore, we invite you to submit a revised version of the manuscript that addresses the points raised during the review process.

We look forward to receiving your revised manuscript.

Kind regards,

Branislav T. Šiler, Ph.D.

Academic Editor

PLOS ONE

Journal Requirements:

**Additional Editor Comments:**

**I believe the authors failed to spot my comments in the previous review round as they remained unaddressed. Please check under *Additional Editor Comments.***

---

## [Author Response · Author response to Decision Letter 1]

17 Apr 2024

Please see the attached 'Response to Reviewers' document.

---

## [Editor Report · Decision Letter 2]

19 Apr 2024

Long-term stability and the physical and chemical factors predictive for antimicrobial activity in Australian honey

PONE-D-24-05003R2

Dear Dr. Fernandes,

We’re pleased to inform you that your manuscript has been judged scientifically suitable for publication and will be formally accepted for publication once it meets all outstanding technical requirements.

Kind regards,

Branislav T. Šiler, Ph.D.

Academic Editor

PLOS ONE
---

## [Editor Report · Acceptance letter]

2 May 2024

PONE-D-24-05003R2 

PLOS ONE

Dear Dr. Fernandes, 

I'm pleased to inform you that your manuscript has been deemed suitable for publication in PLOS ONE. Congratulations! Your manuscript is now being handed over to our production team.

Kind regards, 

on behalf of

Dr. Branislav T. Šiler 

Academic Editor

PLOS ONE